# Modulation of p75^NTR^ on Mesenchymal Stem Cells Increases Their Vascular Protection in Retinal Ischemia-Reperfusion Mouse Model

**DOI:** 10.3390/ijms22020829

**Published:** 2021-01-15

**Authors:** Sally L. Elshaer, Hang-soo Park, Laura Pearson, William D. Hill, Frank M. Longo, Azza B. El-Remessy

**Affiliations:** 1Augusta Biomedical Research Corporation, Charlie Norwood VA Medical Center, Augusta, GA 30901, USA; Dr_s_elshaer@mans.edu.eg (S.L.E.); laura.pearson1@va.gov (L.P.); hillwi@musc.edu (W.D.H.); 2Department of Pharmacology and Toxicology, Faculty of Pharmacy, Mansoura University, Mansoura 35516, Egypt; 3Department of Surgery, University of Illinois at Chicago, Chicago, IL 60612, USA; hspark06@bsd.uchicago.edu; 4Department of Obstetrics and Gynecology, University of Chicago, Chicago, IL 60637, USA; 5Department of Pathology and Laboratory Medicine, Medical University of South Carolina, Charleston, SC 29403, USA; 6Ralph H. Johnson Veterans Affairs Medical Center, Charleston, SC 29403, USA; 7Department of Neurology and Neurological Sciences, Stanford University, Palo Alto, CA 94304, USA; flongo@stanford.edu; 8Department of the Pharmacy, Doctors Hospital of Augusta, Augusta, GA 30909, USA

**Keywords:** mesenchymal stem cells, p75^NTR^, ischemic retinopathy, angiogenesis, survival, NGF, VEGF, SDF-1, ischemia/reperfusion, visual assessment

## Abstract

Mesenchymal stem cells (MSCs) are a promising therapy to improve vascular repair, yet their role in ischemic retinopathy is not fully understood. The aim of this study is to investigate the impact of modulating the neurotrophin receptor; p75^NTR^ on the vascular protection of MSCs in an acute model of retinal ischemia/reperfusion (I/R). Wild type (WT) and p75^NTR-/-^ mice were subjected to I/R injury by increasing intra-ocular pressure to 120 mmHg for 45 min, followed by perfusion. Murine GFP-labeled MSCs (100,000 cells/eye) were injected intravitreally 2 days post-I/R and vascular homing was assessed 1 week later. Acellular capillaries were counted using trypsin digest 10-days post-I/R. In vitro, MSC-p75^NTR^ was modulated either genetically using siRNA or pharmacologically using the p75^NTR^ modulator; LM11A-31, and conditioned media were co-cultured with human retinal endothelial cells (HREs) to examine the angiogenic response. Finally, visual function in mice undergoing retinal I/R and receiving LM11A-31 was assessed by visual-clue water-maze test. I/R significantly increased the number of acellular capillaries (3.2-Fold) in WT retinas, which was partially ameliorated in p75^NTR-/-^ retinas. GFP-MSCs were successfully incorporated and engrafted into retinal vasculature 1 week post injection and normalized the number of acellular capillaries in p75^NTR-/-^ retinas, yet ischemic WT retinas maintained a 2-Fold increase. Silencing p75^NTR^ on GFP-MSCs coincided with a higher number of cells homing to the ischemic WT retinal vasculature and normalized the number of acellular capillaries when compared to ischemic WT retinas receiving scrambled-GFP-MSCs. In vitro, silencing p75^NTR^-MSCs enhanced their secretome, as evidenced by significant increases in SDF-1, VEGF and NGF release in MSCs conditioned medium; improved paracrine angiogenic response in HREs, where HREs showed enhanced migration (1.4-Fold) and tube formation (2-Fold) compared to controls. In parallel, modulating MSCs-p75^NTR^ using LM11A-31 resulted in a similar improvement in MSCs secretome and the enhanced paracrine angiogenic potential of HREs. Further, intervention with LM11A-31 significantly mitigated the decline in visual acuity post retinal I/R injury. In conclusion, p75^NTR^ modulation can potentiate the therapeutic potential of MSCs to harness vascular repair in ischemic retinopathy diseases.

## 1. Introduction

Vision loss associated with ischemic diseases, such as retinopathy of prematurity, diabetic retinopathy and optic neuropathy, are often due to the growth of malfunctional retinal capillaries, in response to retinal ischemia [1,2]. Significant progress has been made in combating abnormal vascular proliferation, including laser photocoagulation, anti-angiogenic vascular endothelial growth factor (VEGF) inhibitors and angiostatic steroids (reviewed in [3]). Nevertheless, these therapeutic approaches fail to treat ischemic areas with a large degree of tissue injury in addition to local and systemic complications associated with their use [4]. Thus, there is an urgent need to develop new therapeutic alternatives to combat early retinal ischemia and thus diminish subsequent destructive neovascularization.

Cell therapy in general and mesenchymal stem cells (MSCs) can be an attractive candidate for retinal regeneration. MSCs are multipotent stem cells present in adult marrow and have the potential to differentiate into lineages of mesenchymal tissues, including bone, cartilage, fat, tendon, muscle and marrow stroma [5]. One advantage of adult MSCs is that they can be isolated from the bone marrow, peripheral blood, or adipose tissue of a subject in reasonable quantity; however, the in vitro expansion of MSCs, a lengthy process, prior to administration, can result in a loss of stemness and can be hampered by the age and the disease state of the patient (reviewed in [6]). MSCs are perceived as “immune-privileged” cells and as such are widely used clinically in allogeneic settings [7,8]. MSCs are home to sites of inflammation where they secrete a variety of soluble factors, including growth factors, cytokines, and chemokines [9]. While the secretion of paracrine factors by MSCs is well-established as a reparative mechanism, the engrafting of MSCs to injured tissues has not been proven to be a perquisite for the reparative action [10,11]. Animal studies demonstrated that the subretinal transplantation of MSCs delays retinal neurodegeneration and preserves neural retinal function [12,13,14]; however, the vascular protection of MSCs transplanted in the ischemic retina has not been fully elucidated.

The p75^NTR^, a common receptor for all neurotrophins along with their precursor forms, can exert multiple functions, including cell survival, death or angiogenesis, according to cell context (reviewed in [15]). Among increasing prospective stem cells markers, the p75^NTR^ receptor, also known as CD271, enriches several progenitor/stem cell subtypes [16]. P75^NTR^ was first identified as a genuine neural crest stem cell marker [17]. Since then, it has been widely used to isolate putative stem cells from neural crest-derived tissues. The p75^NTR^ was shown to directly inhibit the differentiation of MSCS into multiple cell types [18]. Several studies reported a causal relationship of p75^NTR^ in mediating retinal inflammation, barrier dysfunction and development of acellular capillaries [19,20,21,22,23,24]. Nonetheless, p75^NTR^ expression and function in MSCs biology, as well as its underlying mechanisms, have not been fully studied. In the current study, we attempted to evaluate the impact of modulating p75^NTR^ expression or activity on the surface of MSCs in increasing their vascular homing and repair in ischemic retina using an ischemia/reperfusion (I/R) mouse model. Further, the protective action of pharmacological inhibition of p75^NTR^ on improving visual acuity post I/R injury was investigated.

## 2. Results

### 2.1. Deletion of p75^NTR^ Prevented Retinal Capillary Degeneration in Ischemia/Reperfusion Model

Our previous work showed that the genetic deletion of p75^NTR^ prevented diabetes-induced degeneration of retinal capillaries identified by the presence of acellular capillaries [21]. Based on this similarity between ischemia/reperfusion (I/R) and diabetic retinopathy [19], we examined the vascular protective effects of p75^NTR^ genetic deletion in acute retinal ischemic/reperfusion (I/R) injury. Trypsin-digested retinas showed that I/R resulted in a significant increase in the number of acellular capillaries (3-Fold) in WT mice, compared to their sham-operated controls, whereas the sham group showed a mean of 2.51 ± 0.5 acellular capillaries, as compared to the I/R group, showing a mean of 7.06 ± 0.83 acellular capillaries/field (Figure 1A,C). The deletion of the p75^NTR^ receptor significantly reduced the number of acellular capillaries in the ischemic retina by 35%, compared to the ischemic WT retinas. Two-way ANOVA showed the significant effects of disease state (I/R) and gene deletion. There was no significant difference between p75^NTR-/-^ in ischemic retinas when compared to the p75^NTR-/-^ sham group (Figure 2B,C).

### 2.2. Mesenchymal Stem Cells (MSCs) Are Engrafted to Retina Capillaries and Improve Vascular Protection in Both WT and p75^NTR-/-^ Mice

MSCs are widely used; however, whether the incorporation and engrafting of MSCs to injured tissues is required to exert their vascular protection remains to be elucidated. We elected to utilize GFP-labelled MSCs that could be traced post intravitreal injection. I/R was performed and, two-days later, GFP-MSCs were intravitreally injected into WT ischemic retinas. The homing and integration of MSCs into ischemic vasculature were followed after an additional 3 days (Figure 2A) and 1 week (Figure 2B) post injection. Isolectin GS-stained WT retinal flat mounts showed the presence of GFP-labeled MSCs (green) as satellite-shaped within the capillary network (red), with no integration (Figure 2A). On the other hand, at one week post intravitreal injection, GFP-MSCs showed engrafting and incorporation (yellow) within retinal capillaries (Figure 2B). The injection of MSCs enhanced vascular protection against the formation of acellular capillaries following retinal ischemia both in WT and p75^NTR-/-^ retinas (Figure 2C,D). WT ischemic retinas that received MSCs still showed a significant 2-Fold increase in the count of acellular capillaries (3.0 ± 0.29), as compared to the WT sham group (1.51 ± 0.17). In contrast, the ischemic p75^NTR-/-^ retinas that received MSCs showed a similar count for acellular capillaries (1.76 ± 0.31) to the p75^NTR-/-^ sham group (1.56 ± 0.26) and was significantly lower than the count for WT ischemic retinas (3.0 ± 0.29, Figure 2C,D).

### 2.3. Silencing p75^NTR^ Expression in MSCs Increased Their Homing and Vascular Protection in Ischemic Retinas

Since the p75^NTR^ has been shown to possibly inhibit differentiation of MSCS into multiple cell types, we examined the impact of silencing p75^NTR^ on MSCs homing and engrafting into ischemic retinas. Optimization experiments showed that 100 nM of siRNA against p75^NTR^ was as effective as 300 nM of siRNA to reduce p75^NTR^ mRNA expression (Appendix A). To assess the change in vascular homing upon knocking-down p75^NTR^ expression, the Scr- or Si-100-treated GFP-labeled MSCs were injected intravitreally, 2 days post I/R injury in WT mice (Figure 3). Isolectin GS-stained retinal flat mounts showed enhanced recruitment and homing of GFP-labeled MSCs (green) into WT-retinal capillaries (red) after silencing p75^NTR^ expression compared to scrambled cells (Figure 3A). Trypsin-digestion of ischemic WT retinas that received scrambled-p75^NTR^-MSCs still showed a significant 3.3-Fold increase in the number of acellular capillaries (4.55 ± 0.24) when compared to their sham-group (1.27 ± 0.49, Figure 3B,C). Meanwhile, knocking down p75^NTR^ expression on MSCs resulted in almost complete vascular protection of ischemic WT retinas, where there was no significant difference between ischemic group and their shams receiving siRNA-MSC (1.95 ± 0.18 versus 1.65 ± 0.38). The number of acellular capillaries in ischemic retinas receiving siRNA-p75^NTR^-MSCs significantly decreased by ~58%, when compared to ischemic retinas receiving scrambled-p75^NTR^-MSCs; Figure 3B,C).

### 2.4. Silencing p75^NTR^ Expression on MSCs Improves Their Secretome

Secretion of paracrine factors by MSCs is well-established as their primary reparative mechanism [10,11]. Thus, we attempted to characterize the impact of modulating p75^NTR^ on the MSCs-secretome. As shown in Figure 4, silencing p75^NTR^ expression on MSCs significantly increased the protein expression of VEGF (6-Fold), SDF-1 α (1.8-Fold) and NGF (5.8-Fold) in MSCs conditioned medium (CM) when compared to control CM.

### 2.5. Silencing p75^NTR^ Expression on MSCs Improves Paracrine Effect on HREs

Since an improved MSC-secretome was observed by silencing p75^NTR^ expression on MSCs (Figure 4), we examined the paracrine effect of MSCs CM on HREs. As shown in Figure 5, the treatment of HREs with CM of p75^NTR^-silenced MSCs increased gene expression of the anti-apoptotic markers *VEGF-A* by 5-Fold (5.01 ± 0.24 versus 1.02 ± 0.13 for controls, Figure 5A), *Akt* by ~10-Fold (12.30 ± 3.6 versus 1.18± 0.39 for controls; Figure 5B), *Bcl-2* by ~6.8-Fold (6.86 ± 0.33 versus 1.01 ± 0.11 for controls; Figure 5C). For the apoptotic markers, silencing p75^NTR^ resulted in increases in *Bax* expression by ~9.6-Fold (9.65 ± 0.21 versus 1.01 ± 0.09 for controls; Figure 5D), however there was no change in *p53* (1.27 ± 0.54 vs. 1.35 ± 0.98; Figure 5E) or *caspase-3* expression (1.1 ± 0.4 vs. 1.4 ± 1.8; Figure 5F).

### 2.6. Silencing p75^NTR^ Expression on MSCs Enhanced Angiogenic Response in HREs

The angiogenic behavior of HREs showed significant improvement upon treatment with CM of p75^NTR^-silenced MSCs, where HREs migration significantly increased by 1.4-Fold after 12 h of treatment (Figure 6A,B). In addition, the ability of HREs to form tubes in vitro was significantly enhanced following 24 h of treatment, in terms of tube length that showed an increase of 1.5-Fold and the number of junction points that showed an increase of 2-Fold (Figure 6C–E).

### 2.7. Modulating p75^NTR^ on MSCs Using LM11A-31 Improved Their Secretome and Improved Paracrine Effect in HREs

LM11A-31, the pharmacologic modulator of p75^NTR^ receptor, showed similar results to p75^NTR^ genetic silencing in improving the secretome of MSCs. As shown in Figure 7A–D, the treatment of MSCs with LM11A-31 increased the release of SDF-1a, VEGF-A and NGF in their CM. SDF-1a increased by ~14.9-Fold, VEGF-A increased by 1.6-Fold and NGF increased by ~4-Fold when compared to control CM. As shown in Figure 7E–I, the treatment of HREs with conditioned medium of LM11A-31-treated MSCs increased the gene expression of *VEGF-A* by ~3.5-Fold (3.58 ± 0.58 versus 1.017 ± 0.129 for controls; Figure 7E), *Akt* by ~2.7-Fold (2.68 ± 0.53 versus 1.0 ± 0.06 for controls; Figure 7F), *Bcl-2* by ~2.5-Fold (2.45 ± 0.24 versus 1.0 ± 0.04 for controls; Figure 7G). For some of the apoptotic markers, CM of LM11A-31-treated MSCs increased gene expression of *Bax* by ~1.9-Fold (1.88 ± 0.105 versus 1.0 ± 0.055 for controls; Figure 7H), but there was no change in the expression of *p53* (3.7 ± 3.4 vs. 5.7 ± 4.2; Figure 7I).

### 2.8. Conditioned Medium of LM11A-31-Treated MSCs Enhanced Angiogenic Response in HREs 

CM of p75^NTR^-modulated MSCs using LM11A-31 showed similar enhanced angiogenic response in HREs (Figure 8) to our observation using p75^NTR^-siRNA (Figure 6). HREs migration significantly increased by 1.34-Fold after 12 h of treatment (Figure 8A,B). Further, the ability of HREs to form tubes in vitro was significantly enhanced following 24 h of treatment, evident by 1.4-Fold increase in tube length and 1.5-Fold increase in number of junction points when compared to controls (Figure 8C–E).

### 2.9. Modulating p75^NTR^ Using LM11A-31 Improves Visual Acuity in WT Mice Post I/R Injury

As shown in Figure 9, mice subjected to I/R injury showed deteriorated visual function in the visual-clue water-maze test, where time to reach the platform was significantly increased in mice that underwent I/R injury after 6 days, 9 days, 12 days and 15 days post I/R, an effect that was significantly ameliorated in mice receiving LM11A-31 at all tested time points.

## 3. Discussion

Retinal ischemia is a common underlying pathology for multiple retinal diseases, including diabetic retinopathy, retinopathy of prematurity, traumatic optic neuropathy and acute closed-angle glaucoma [1,2]. Vascular cell death and development of acellular capillaries are well-accepted surrogate markers for retinal ischemia. Stem cell therapy, in general, and mesenchymal stem cells (MSCs), in particular, have demonstrated hopeful results to improve ischemic and neuronal disorders. We and others demonstrated that increases in the neurotrophin receptor p75^NTR^ are closely linked to retinal inflammation, barrier dysfunction and the development of acellular capillaries [19,20,21,22,23,24]. Here, we demonstrate the vascular protective effects of modulating p75^NTR^ and MSCs injection using an acute model of retinal ischemia/reperfusion in mice. The main findings of the current study are: (1) the vascular protective effects of p75^NTR^ deletion against retinal ischemia is enhanced by MSCs injection; (2) silencing p75^NTR^ expression on MSCs enhanced their vascular homing and potentiated the vascular protection against I/R-injury; (3) modulating p75^NTR^ on MSCs using siRNA or pharmacologically using LM11A-31 enriched their secretome with trophic and angiogenic factors, such as NGF, SDF-1, and VEGF; (4) conditioned medium of p75^NTR^-modulated MSCs stimulated the survival and angiogenic behavior of retinal endothelial cells; (5) further intervention with LM11A-31 significantly improved decline in visual acuity post retinal ischemic injury.

P75^NTR^ signal transduction pathways are extremely variable because they are dependent on cell type, cell differentiation status, neurotrophin binding, interacting transmembrane co-receptor as well as the availability of intra-cellular adaptor molecule [25]. This leads to divergent cellular responses, including progenitor differentiation [26], cell survival [27], apoptosis [20,21,28], and cell migration and invasion [22,29]. Our findings showed that transient exposure to I/R resulted in a significant increase in acellular capillaries formation in WT mice that was ameliorated in p75^NTR-/-^ mice (Figure 1). In support, prior study showed that vascular cell death and the development of acellular capillaries are associated with increases in the expression of p75^NTR^ and its co-receptor sortilin post retinal I/R-injury [24]. The vascular protection associated with p75^NTR^ modulation have been attributed to the inhibition of inflammation and cell death signaling in capillary cells [20,21,22,23,24,30]. 

Capillary degeneration associated with p75^NTR^ signaling may be due to failure of endothelial hematopoietic stem cells to maintain normal retinal vasculature or to repair damaged vasculature [31]. The P75^NTR^ receptor, also known as CD271, enriches several progenitor/stem cells subtypes, including MSCs [16]. P75^NTR^ is involved in MSCs differentiation where expression of p75^NTR^ was shown to rapidly down-regulate upon differentiation of MSCs in vitro [32]. Moreover, p75^NTR^ could directly hinder the differentiation of MSCs through the inhibition of transcription factors, including Runx2 and OSX, which are essential for osteoblast differentiation and for the expression of chondrogenesis marker; Sox9 and the myogenic marker, Myf5 [18]. The transplantation of bone marrow-derived MSCs was shown to rescue photoreceptor cells in dystrophic retina of rhodopsin knockout mice, suggesting a therapeutic benefit in retinitis pigmentosa [33]. The neuroprotection of MSCs has been reported in rat retina subjected to I/R, where injection of MSCs preserved number of RGCs compared to controls [34]. While MSCs-associated neuroprotection has been demonstrated, our study explored the MSCs-mediated vascular protection in ischemic retina. The intravitreal injection of MSCs 2-days post I/R injury exerted vascular protection in both WT and p75^NTR-/-^ mice with enhanced homing and integration into retinal capillaries (Figure 2). Interestingly, silencing p75^NTR^ expression on the surface of MSCs resulted in increased vascular homing of MSCs and potentiated the vascular protection evident by significant decreases in the number of acellular capillaries (Figure 3).

To explore the possible behavior of MSCs post injection and its interaction with retinal microvasculature, we elected to inject a large number of MSCs (100,000 cells/eye). It was interesting to observe different behavior at different time-points (Figure 2A,B). At the 3-day time-point, some of the MSCs morphed into a satellite shape and remained green, as they were not with the same plane of retinal capillaries (Figure 2A: left). Other MSCs morphed into yellow, ameboid-shaped cells that co-stained with isolectin (Figure 2A: right). Interestingly, 1-week post injection, some MSCs integrated into perivascular and wrapped around retinal capillaries (red) but remained green (Figure 2B: Left), while others appeared yellow and retained the ameboid shape and isolectin co-staining (Figure 2B: Right). These observations confirm the recruitment and homing of MSCs to the ischemic retina vasculature. The co-staining of GFP-MSCs with isolectin (yellow) might suggest possible interaction with microglia and modulation of proinflammatory phenotype, as recently shown [35]. The perivascular appearance raises the possibility of MSCs being differentiated into perivascular cells, such as pericytes. Pericytes have mesenchymal origin and are traditionally recruited to support endothelial tubes [36]. Our findings lend further support to the potential differentiation of injected MSCs into pericytes, as illustrated previously [37,38]. However, these interesting observations warrant further detailed investigation beyond the scope of the current study.

To investigate the underlying mechanism behind the augmented vascular protection observed after silencing p75^NTR^ expression on MSCs, their CM were examined for secreted trophic and growth factors. The silencing of MSCs-p75^NTR^ enriched their CM with VEGF, NGF and SDF-1α (Figure 4), which coincided with enhanced HREs survival (Figure 5) and paracrine angiogenic response (Figure 6). We have also seen tendency for increased gene expression of SDF-1α receptors, *CXCR-4* and *CXCR-7* (Appendix A). SDF-1 is a classical chemo-attractant stimulus of endothelial progenitor cells to endothelial lineage that is known to upregulate following ischemic insult [39]. In agreement with our findings, MSCs overexpressing SDF-1 were shown to have increased expression of VEGF, Akt and eNOS. In vivo, they enhanced angiogenesis and improved heart function in an experimental myocardial infarction rat model [40]. In addition, SDF-1 was shown to mediate the regenerative effects of MSCs, where knocking-down MSCs-SDF-1 significantly reduced their beneficial effects on alveolarization, angiogenesis, and inflammation in a rodent model of bronchopulmonary dysplasia [41]. It is well-known that the transendothelial migration of MSCs is enhanced through the SDF-1/CXCR4 axis [42]. The SDF-1/CXCR4 axis has been shown to improve the recruitment and hematopoietic reconstitution of bone marrow-derived MSCs in aplastic anemia [43]. Recently, atypical chemokine receptor ACKR3, previously known as CXCR7, was also shown to promote the proliferation and migration of MSCs, as well as their secreted SDF-1, improving their treatment effectiveness [44]. Moreover, SDF-1/CXCR-7 was reported to enhance the migration of MSCs in a transient cerebral I/R rat hippocampus model [45]. Thus, enhanced in vivo vascular integration and protection, as well as in vitro paracrine angiogenic potential of HREs induced by silencing MSCs-p75^NTR^ can be attributed, at least in part, to SDF-1/CXCR-4 and-7 signaling axis. VEGF and Akt are known regulators of endothelial cell survival and angiogenesis [46]. In vitro, the treatment of HREs with enriched CM of p75^NTR^-silenced MSCs improved HREs survival, evident by the increased gene expression of *VEGF* and *Akt* (Figure 5) as well as their angiogenic potential in vitro (Figure 6). In support of our observation, a prior study reported that MSCs-secreted VEGF mediates the differentiation of endothelial progenitor cells into endothelial cells via paracrine mechanisms [47]. MSCs-secreted VEGF was also shown to play a part in underlying paracrine and neuroprotective action of MSCs [48]. NGF is another potential trophic factor that can contribute to enhanced reparative potential of p75^NTR^-silenced MSCs. We and others have demonstrated that modulation of p75^NTR^ enhances trophic support by decreasing the levels of proNGF and proBDNF and increasing the mature form of NGF and BDNF [22,30,49]. Our findings lend further support to prior reports, showing that NGF enhanced angiogenesis and tube formation of MSCs in PI3K/Akt-dependent manner [50]. Additionally, NGF was shown to promote endothelial progenitor cell-mediated angiogenic responses via SDF-1/CXCR-4, with minimal effect on resident retinal endothelial cells [51].

Of note, we explored changes in the gene expression of both survival and apoptotic markers in HREs upon treatment with the CM of p75^NTR^-silenced MSCs. Conventionally, the Bcl-2/Bax axis regulates mitochondrial-induced apoptosis in endothelial cells through the reciprocation of the anti-apoptotic proteins; Bcl-2 and the pro-apoptotic protein; Bax, which might result in a neutral net effect [52]. Remarkably, we observed marked increases in the expression of both *Bcl-2* and *Bax* in HREs treated with CM of p75^NTR^-silenced MSCs. Although the pro-apoptotic marker *BAX* was increased, other pro-apoptotic markers, such as *p53* and *caspase-3* transcripts, did not show significant difference from the controls (Figure 5). In light of other increased survival and angiogenic factors, such as VEGF and NGF, it is conceivable that the CM of p75^NTR^-silenced MSCs exerted an anti-apoptotic and proangiogenic effect on HREs.

In parallel to genetic silencing of MSCs-p75^NTR^ data, pharmacological inhibition of MSCs-p75^NTR^ using LM11A-31 enriched MSCs secretome with SDF-1α, NGF and VEGFA and augmented the paracrine angiogenic behavior on endothelial cells in vitro (Figure 7 and Figure 8). LM11A-31 (2-amino-3-methyl-pentanoic acid [2-morpholin-4-yl-ethyl]-amide) is a water-soluble isoleucine derivative. The specificity of LM11A-31 to p75^NTR^ has been well-documented by the complete loss of its protective activity in cultures of p75^NTR−/−^ neurons, [53,54]. We and others have demonstrated the anti-inflammatory and neuro- and vascular protective effects of LM11A-31 [30,53,55]. However, we believe that this is the first report showing the impact of LM11A-31 on stem cells and its protective effects on visual acuity in a mouse-model of the ischemic retina (Figure 9). At the dose used (50 mg/kg), LM11A-31 has been shown to cross the blood–brain barrier following oral administration [55]. Of note, LM11A-31 was approved by the US Food and Drug Administration as an investigational new drug, successfully completed a phase I safety trial in normal young and elderly individuals and is currently undergoing evaluation in a phase 2a exploratory endpoint trial in Alzheimer’s disease (ClinTrials.gov registration no. NCT03069014).

Among stem cells including embryonic, neural and adult stem cells, MSCs gained interest because they can be obtained from the patients’ bone marrow in quantities appropriate for clinical application. Since the expansion of MSCs from each patient can be possibly hampered by the disease state, allogenic cell therapy will be the way to go. Here, we provide evidence that combination of p75^NTR^ modulation strategy and MSCs injection can serve as novel therapeutic approach to rescue visual function in ischemic retinal diseases with superior therapeutic potential than solely using MSCs. The underlying mechanism can involve augmented SDF-1, VEGF and NGF signaling pathways. The current findings support, for the first time, the therapeutic potential of the orally bioavailable compound, LM11A-31 for ischemic retinal diseases. While we were able to observe homing and integration of MSCs into retina vasculature and correlate that process with vascular protection post-ischemic injury, the nature and extent of differentiation warrants further investigation. Future studies are needed to fully understand the role of the p75^NTR^ receptor in MSCs physiology, as well as its precise underlying mechanisms in regulating MSCs neurogenesis and angiogenesis.

## 4. Materials and Methods

### 4.1. Animals

All animal experiments were conducted in agreement with Association for Research in Vision and Ophthalmology statement for use of animals in ophthalmic and vision research, and Charlie Norwood VA Medical Center Animal Care and Use Committee (ACORP#16–01–088). The p75^NTR^, B6.129S4Ngfrtm1Jae /J (p75^NTR-/-^, exon III knockout mice were obtained from Jackson Laboratories (Bar Harbor, Maine, USA) and crossed with C57BL6-J mice (Jackson Laboratories). These mice were crossed and back-crossed to establish a colony of homozygous p75^NTR-/-^ and WT breeders that produced the mice used in the current study.

### 4.2. Retinal Ischemia/Reperfusion

For surgeries, mice were anesthetized with intraperitoneal ketamine (50 mg/kg; Hospira, Inc., Lake Forest, IL, USA) and xylazine (10 mg/kg; Akorn, Decatur, IL, USA). Retinal ischemia/reperfusion (I/R) was performed as described previously [56]. Briefly, pupils were dilated with 1% Atropine Sulfate (Akorn, Inc., Lake Forest, IL, USA). The anterior chamber was cannulated with a 32-gauge needle attached to a line from a saline reservoir at a height calibrated to yield 120 mmHg. The intraocular pressure (IOP) was elevated to 120 mmHg for 45–60 min. I/R injury and choroidal non-perfusion was evident by whitening of the anterior segment of the globe and blanching of the episcleral veins. During infusion, topical anesthesia (0.5% tetracaine HCL, Bausch & Lomb, Bridgewater Township, NJ, USA) was applied to the cornea. After ischemia, the needle was immediately withdrawn, allowing for rapid reperfusion, IOP was normalized, and reflow of the retinal vasculature was confirmed by observation of the episcleral veins. Topical antibiotic (Neo-Polycin, Perrigo, Allegan, MI, USA) was applied to the cornea to minimize infection. I/R injury was performed in one eye with the other undergoing sham surgery, in which the needle was inserted into the anterior chamber without elevating the IOP. Mice were sacrificed 10 days post I/R and eyes were processed.

### 4.3. Mesenchymal Stem Cells (MSCs) Culture

GFP-labeled mouse MSCs were obtained as a kind gift from Dr. William D. Hill (MUSC, Charleston, SC, USA) [57] and used between passages 5–9. MSCs cells were cultured in high glucose DMEM (4.5 g/L D-glucose, Thermo-Fisher Grand Island, NY, USA) supplemented with 10% FBS and 1% penicillin/streptomycin at 37 °C and 5% CO_2_. Knocking down p75^NTR^ expression was performed according to manufacturer’s instructions (Santa Cruz Biotechnology Inc., Dallas, TX, USA). Briefly, MSCs were shifted to antibiotic-free medium over night when 60–80% confluent. Cells were transfected with Si-RNA against p75^NTR^ receptor (sc-37268) or scrambled (sc-37007) with the aid of lipofectamine transfection reagent (sc-29528) for 6 h. DMEM medium containing 2X of FBS and antibiotic was added on top of pure transfection medium for an extra 12 h. Next, transfection medium was removed and cells were allowed to recover in 1X DMEM medium for 6 h. For in vivo studies, cells were collected in sterile PBS (50,000 cells/μL) for intravitreal injection 48 h after I/R. For additional set of experiments, MSCs were treated with the p75^NTR^ modulator (LM11A-31, 200 nM), provided by Dr. Frank Longo, Stanford University. Confluent MSCs (80% confluent) were treated with LM11A-31 or its vehicle for 12 h. Then, condition media were collected as well as cell lysates using mirVANA^TM^ PARIS^TM^ Kit (Ambion Inc., Austin, TX, USA), according to manufacturer’s instructions.

### 4.4. Intravitreal Injection of MSCs

Mice were anesthetized by intraperitoneal injection of ketamine (50 mg/kg, Hospira,) and xylazine (10 mg/kg, Akorn) mixture and complete anesthesia was confirmed by loss of reflex to sharp paw pinch. MSCs (100,000 cells/2 μL sterile PBS/ eye) were injected intravitreally 48 h after I/R using a Hamilton syringe with a 33-gauge glass capillary.

### 4.5. Vascular Localization of MSCs

3 or 7 days post intravitreal injection of MSCs, mice were euthanized in CO_2_ chamber (2% flow rate for 5 min), followed by cervical dislocation. Eyes were enucleated and fixed in 2% paraformaldehyde overnight. Retinas were dissected and permeabilized for 15 min with 0.3% Triton X-PBS then stained overnight at 4 °C with isolectin B4; biotinylated griffonia (bandeiraea) simplicifolia lectin I (GSL I, BSL I), (Vector Labs, Burlingame, CA, USA; 1% in 5% normal goat serum in 0.3% Triton X-PBS), followed by incubation with secondary antibody; Texas red^®^ avidin D (Vector labs; 0.5% in 5% normal goat serum in 0.3% Triton X-PBS). Lectin-stained retinas were flat-mounted onto Superfrost/Plus microscope slides (Fisher Scientific, Waltham, MA, USA) with the photoreceptor side facing down and imbedded in Vectashield mounting media for fluorescence (Vector Labs). Slides were photo-micro-graphed at 40x using a Zeiss Axio Observer Z1 (Carl Zeiss Vision Inc.,Thornwood, NY, USA).

### 4.6. Isolation of Retinal Vasculature and Determination of Aacellular Capillaries

The retinal vasculature was isolated as described previously [58]. Freshly enucleated eyes were fixed with 2% paraformaldehyde overnight. Retinal cups were dissected, washed in PBS, then incubated with 3% Difco-trypsin 250 (BD Biosciences, San Jose, CA, USA) in 25 mmol/l Tris buffer, pH 8, at 37 °C for 1–1.5 h. Vitreous and nonvascular cells were gently removed from the vasculature, which was soaked in several washes of 0.5% Triton X-100 to remove the neuronal retina. Trypsin-digested retinas were stained with periodic acid–Schiff and hematoxylin (PASH). Numbers of acellular capillaries were quantified in six different areas of the mid-retina using bright field microscopy (20×) in a masked manner by two different researchers. Acellular capillaries were identified as capillary-sized blood vessel tubes with no nuclei along their length.

### 4.7. Endothelial Cell Cultures

All human retinal endothelial (HRE) cell studies were in accordance with Association for Research in Vision and Ophthalmology and Charlie Norwood Veterans Affairs Medical Center, research and ethics committee. HREs and supplies were purchased from Cell Systems Corporation (Kirkland, WA, USA) and VEC Technology Inc. (Rensselaer, NY, USA).

### 4.8. Real-Time Quantitative PCR

Retinas samples and MSCs lysates were processed using (mirVANA^TM^ PARIS^TM^ Kit, Ambion) and RNA was purified and quantified as described by the manufacturer’s instructions. A one-step quantitative RT-PCR kit (Invitrogen, Carlsbad, CA, USA) was used to amplify 10 ng retinal mRNA, as described previously [58]. Quantitative PCR was conducted using a StepOnePlus qPCR system (Applied BioSystems, Life Technologies, Waltham, MA, USA). For HREs, RNA isolation was performed using the TRIzol Reagent (Invitrogen, Waltham, MA, USA), according to the manufacturer’s instructions. RNA concentration was quantified by spectrophotometry at 260 nm using Nanodrop 2000 (Thermo Fisher Scientific, Waltham, MA, USA). A total of 1µg of RNA was reverse transcribed to prepare cDNA using the RNA to cDNA EcoDry premix (Takara Bio USA Inc., Mountain View, CA, USA). Real-time PCR was performed using the CFX96 PCR instrument with matched primers (Table 1) and Universal SYBR Green Supermix (Bio-Rad, Hercules, CA, USA). The following PCR parameters were used: initial denaturation cycle at 95 °C for 3 min, followed by 40 amplification cycles at 95 °C for 10 s, 56 °C for 15 s, and 72 °C for 1 min. The results are presented as the fold change in relative gene expression quantified using the delta–delta CT method and normalized to internal controls (18S, *GAPDH*) and expressed relative to controls.

### 4.9. Cell Migration Assay

HREs were grown to confluence and then were wound with a single sterile cell scraper of constant diameter as described previously [59]. Cells were divided into three group, treated with regular media without FBS, and 1:1 mixture of FBS-free regular media and CM of vehicle- or treated-MSCs. Images of wounded areas were taken immediately after adding the treatment using a phase contrast microscope, and baseline were marked under the cell culture dish with blue marker. After 12 h, images of same field were taken and % cell migration was calculated. Each condition was verified in triplicate and was repeated using independent cultures.

### 4.10. Tube Formation Assay

Corning Matrigel Matrix (10 mg protein/mL) Growth Factor Reduced (Corning Inc., Cat. No. 354230, Corning, NY, USA) was used for tube formation assay. Then, 289 μL of cold (4 °C) Matrigel per well was added into chilled 24-well flat-bottom tissue culture plate (USA Scientific Inc., Ocala, FL, USA) and polymerized for 1 h at 37 °C, 5% CO_2_ incubator. HREs were trypsinized and separated into different groups. Each group of cells was centrifuged and cell pellets were resuspended in regular media without FBS, same media with 1:1 mixture of FBS free regular media and CM of vehicle- or treated-MSCs. Then 300 μL of the cell suspension was added onto each well of Matrigel coated 24-well culture plate and incubated at 37 °C. After 24-h, cells-forming tube structures were analyzed by microscopy. Cells in five replicate fields of triplicate wells were digitally photographed with an EVOS microscope (Thermo Fisher Scientific Inc.). The images for Tube formation assay were analyzed using Angiogenesis analyzer for imageJ as previously described [60,61]. An area of 1.8 mm^2^ from each well was analyzed with imageJ ver. 1.53c, and 3 wells were analyzed per group. The software automatically generates the skeletonized trees overlay to highlight junction, branches, and segments (Figure 6C lower panel, Figure 8C lower panel). Master junctions were identified as junctions linking at least three master segments. The total tubular length was calculated as the average of the total tubule length from three wells, five random fields per well.

### 4.11. Visual Acuity Test

Visual acuity was assessed behaviorally by training and testing mice on the "cue" version of the Morris water maze task, as described previously [56]. Mice used for this experiment underwent I/R in both eyes. They were trained for 3 days before I/R and visual acuity was tested at 3, 6, 9, 12 and 15 days post I/R. Animals were placed for 10 s on a platform in a tank of opaque water, 22–25 °C, which was elevated above the water surface (2 cm) and clearly visible from any location in the tank. Subsequently, there were four trials per day for 3 days or until a stable performance plateau was reached. On each trial, animals started from different locations at the periphery of the tank and were allowed to swim to the escape platform. If they did not reach the platform in 60 s, they were gently guided to it by the investigator. They remained on the platform for 10 s. Visual impairment was diagnosed by higher escape time. Treatment groups received oral gavage of LM11A-31 (50 mg/kg/day) every other day, starting 48 h post I/R.

### 4.12. Statistical Analysis

All the data are expressed as mean ± SEM. Differences between 2 groups were detected using un-paired Student T-test. One-way ANOVA was used to assess significant differences between 3 groups. Two-way ANOVA was used to assess interaction between two variables: 2 genes (WT vs. p75^NTR-/-^) X I/R exposure (I/R vs Sham). Tukey–Kramer post-multiple comparisons was used for significant interactions among various groups. Significance for all tests was determined at α = 0.05, Graph-pad Prism, Ver.6 (GraphPad Software, San Diego, CA, USA).

## 5. Patents

A.B.E. and F.M.L. are listed as inventors of patents relating to LM11A-31.

## Figures and Tables

**Figure 1 ijms-22-00829-f001:**
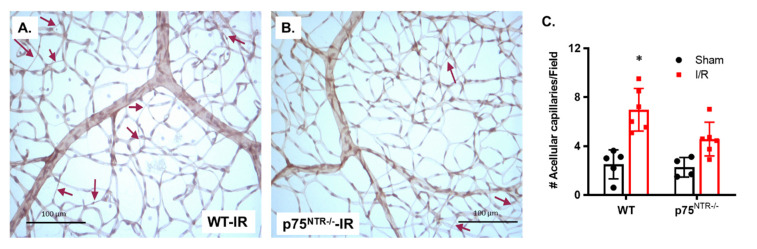
Deletion of p75^NTR^ reduced the development of acellular capillaries in ischemic retinas. (**A**,**B**) Representative trypsin-digested and PASH-stained retinas of WT and p75^NTR-/-^ mice subjected to I/R. Bright field imaging showed an increased number of acellular capillaries (red arrows) in WT that was attenuated in p75^NTR-/-^ mice (20x magnification. (**C**) Statistical analysis using two-way ANOVA showed significant impact of disease condition (ischemia) and gene deletion on the number of acellular capillaries. I/R significantly increased the number of acellular capillaries in WT retinas when compared to shams. This effect was ameliorated in retinas of p75^NTR-/-^ mice (*, significant using two-way ANOVA, *n* = 4–5).

**Figure 2 ijms-22-00829-f002:**
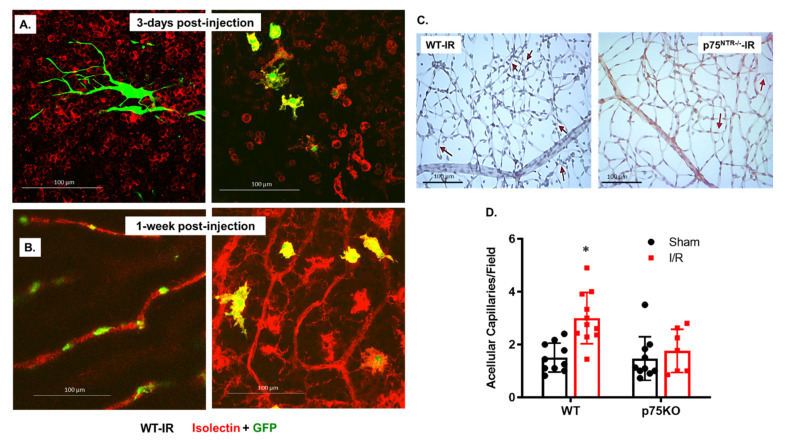
MSCs incorporate to ischemic retinal vasculature and decreased acellular capillaries in WT and p75^NTR-/-^ ischemic retinas. (**A**,**B**) Confocal images of retinal flat mounts stained with isolectin (red), showing incorporation of intravitreally-injected GFP-labeled MSCs (green) into WT retinas subjected to I/R (40x magnification). (**A**) Different regions of flat-mounted retina 3 days post injections showing some MSCs morphed into amoboid/satellite-like shape (green, left panel) and some MSCs picked up isolectin stain (yellow, right panel). (**B**) Different regions of flat-mounted retina 1 week post injection showing, in the left panel, that some of the MSCs appeared perivascular (green) wrapped around retinal capillaries (red) and some other MSCs (right panel) maintained satellite-like shape and co-stained with isolectin (yellow) within retinal capillaries (red), suggesting integration into retinal vasculature. (**C**) Representative trypsin-digested and PASH-stained retinas of WT and p75^NTR-/-^ mice subjected to I/R and receiving GFP-labeled MSCs. Bright field imaging showed decreased count for acellular capillaries (red arrows, 20x magnification). (**D**) Bar graph and statistical analysis for acellular capillaries count in WT and p75^NTR-/-^ retinas subjected to I/R and receiving GFP-labeled MSCs. Ischemic WT retinas still showed a significant increase in the count of acellular capillaries that was completely ameliorated in p75^NTR-/-^ retinas following MSCs injection (*, significant using two-way ANOVA, *n* = 7–11).

**Figure 3 ijms-22-00829-f003:**
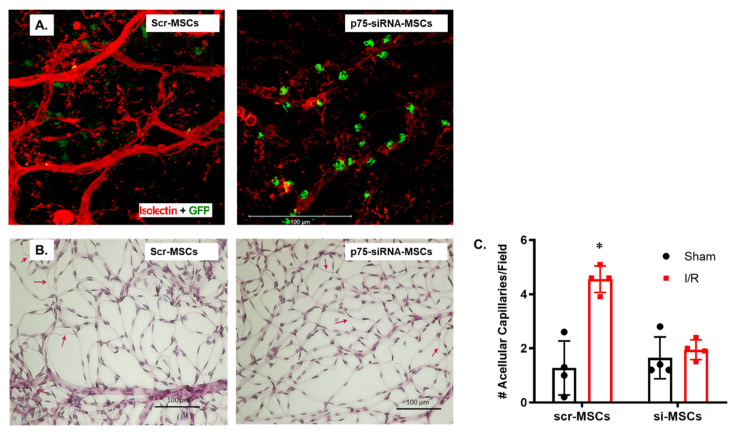
Silencing p75^NTR^ expression in MSCs increased vascular homing and protection in ischemic retinas. (**A**) Confocal images of ischemic WT retinal flat mounts stained with Isolectin-GS (red), showing numerous GFP-labeled MSCs (green) recruited to retinal capillaries (red) after silencing p75^NTR^ expression when compared to scrambled-MSCs (40x magnification). (**B**) Representative trypsin-digested and PASH-stained ischemic WT retinas showing decreased acellular capillaries (red arrows) in ischemic retinas receiving MSCs that do not express p75^NTR^ (20x magnification). (**C**) Bar graph and statistical analysis for acellular capillaries count in ischemic WT retinas receiving Scrambled mRNA-treated or siRNA-treated MSCs against p75^NTR^. Number of acellular capillaries was significantly higher in retinas receiving p75^NTR^-expressing MSCs but not p75^NTR^-knocked down cells (*, significant using two-way ANOVA, *n* = 4).

**Figure 4 ijms-22-00829-f004:**
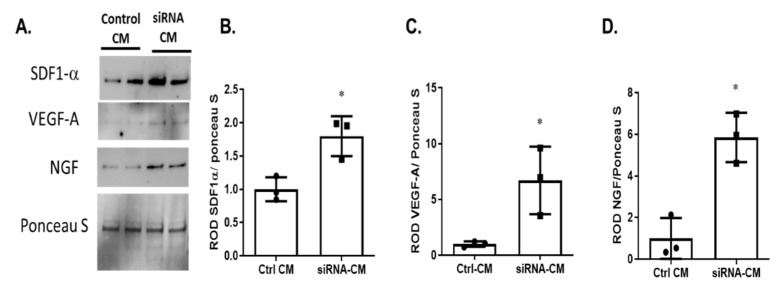
Silencing p75^NTR^ expression improves MSCs-secretome. Representative micrograph of Western blotting (**A**) and statistical analysis (**B**–**D**) of GFP-MSCs CM showed 1.8-Fold increase in SDF-1-a, 6-Fold increase in VEGF-A and 5.8-Fold increase in NGF, upon silencing the expression of p75^NTR^ (* *p* < 0.05, *n* = 3. Significance detected by unpaired student T-test).

**Figure 5 ijms-22-00829-f005:**
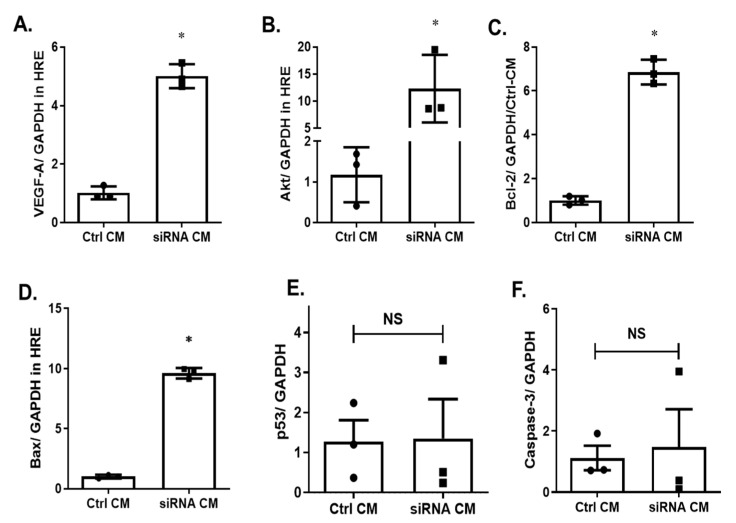
Silencing p75^NTR^ expression on MSCs improves paracrine effect in HREs. RT-PCR analysis showed that treatment of HREs with CM of p75^NTR^-silenced MSCs significantly increased gene expression of angiogenic and survival factor transcripts, including *VEGF-A* (**A**), *Akt* (**B**), *Bcl-2* (**C**) and some of the apoptotic markers including *BAX* (**D**)**,** but no change was observed in *p53* (**E**) or *caspase-3* (**F**) transcripts when compared to HREs treated with CM of control cells. *, significant at *p* < 0.05; NS, not significant. Significance was detected by unpaired student T-test, *n* = 3.

**Figure 6 ijms-22-00829-f006:**
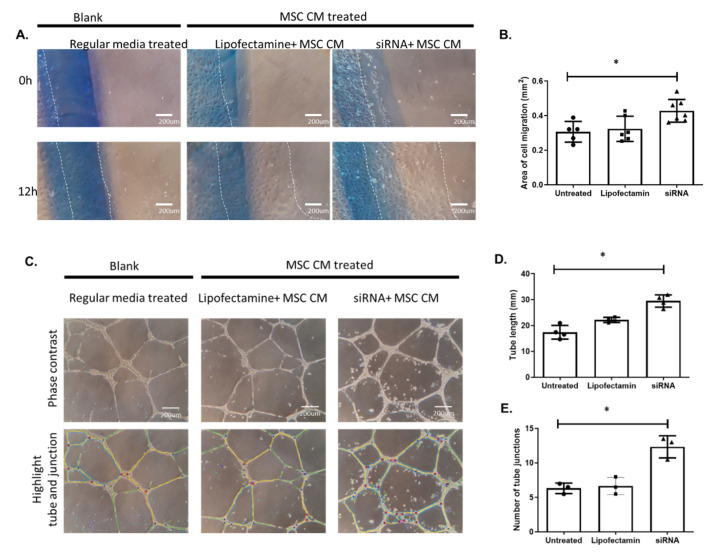
Silencing p75^NTR^ expression on MSCs improves angiogenic response in HREs. (**A**,**B**) Representative images and statistical analysis of HREs migration shows significant increase in HREs migration (distance marked from original plating line marked with blue pen) after 12 h treatment with CM of p75^NTR^-silenced MSCs (*, significance using one-way ANOVA test, *n* = 5–7. (**C**–**E**) Representative phase contrast images (**C**) and statistical analysis for tube length (**D**) and number of tube junction (**E**) formed in vitro by HREs after 24 h treatment with conditioned medium of p75^NTR^-silenced MSCs. Results showed significant increase in tube length and number of tube junctions (*, significance using one-way ANOVA test, *n*= 3).

**Figure 7 ijms-22-00829-f007:**
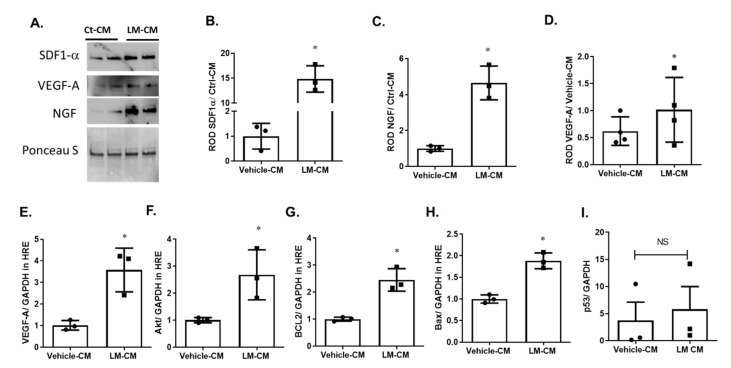
Modulating p75^NTR^ on MSCs using LM11A-31 improved their secretome and improved paracrine effect in HREs. Representative Western blotting (**A**) and statistical analysis of MSCs secretome treated with the pharmacologic modulator of p75^NTR^; LM11A-31. (**B**–**D**) showing ~14.9-Fold increase in SDF-1a (**B**), 1.6-Fold increase in VEGF-A (**C**) and ~4-Fold increase in NGF (**D**). * *p* < 0.05, Significance was detected by unpaired Student t-test, *n* = 3. RT-PCR analysis showed that treatment of HREs with CM of LM11A-31-treated MSCs significantly increased gene expression of the survival factor transcripts *VEGF-A* (**E**), *Akt* (**F**), *Bcl-2* (**G**) and some of the apoptotic markers, such as *Bax* (**H**), but not *p53* (**I**). *, Significant at *p* < 0.05; NS, not significant. Significance was detected by unpaired Student t-test, *n* = 3.

**Figure 8 ijms-22-00829-f008:**
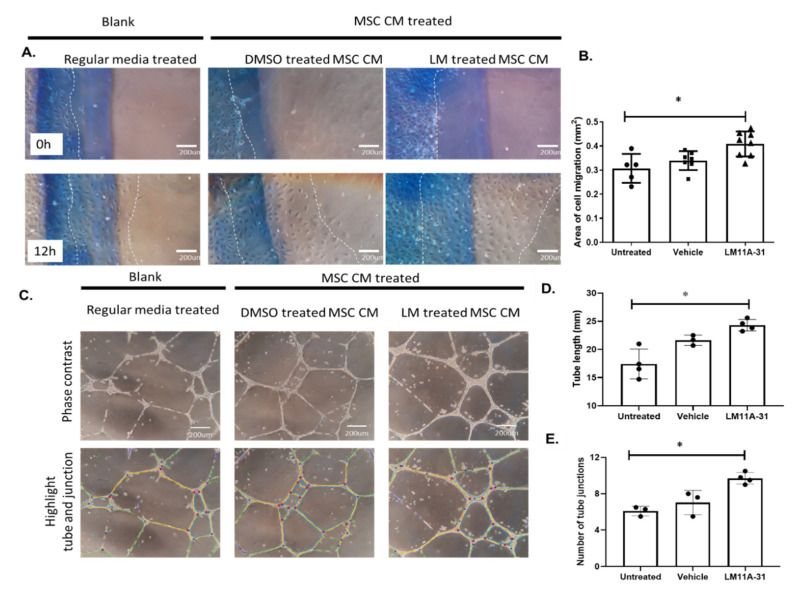
Conditioned medium of LM11A-31-treated MSCs improved angiogenic response in HREs. (**A**,**B**) Representative micrographs and statistical analysis of HREs migration shows a significant increase in HREs migration (distance marked from original plating line marked with blue pen) after 12 h treatment with CM of LM11A-31-modulated MSCs (*, significance using unpaired student t-test, *n* = 5–7). (**C**–**E**) Representative phase contrast images (**C**) and statistical analysis for tube length (**D**) and number of tube junction (**E**) formed in vitro by HREs after 24 h treatment with CM of LM11A-31-modulated MSCs. Results showed a significant increase in tube length and number of tube junctions (*, significance using unpaired student t-test, *n* = 3).

**Figure 9 ijms-22-00829-f009:**
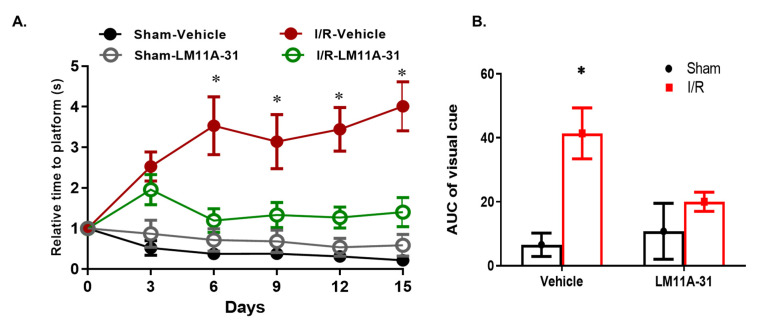
LM11A-31 improve visual acuity in WT mice post I/R injury. (**A**) Follow-up chart of visual-clue water-maze test showing relative time to reach the platform (seconds) for mice undergoing I/R injury and receiving LM11A-31 for up to 15 days post I/R injury. Mice subjected to I/R injury showed significantly longer time to reach the platform starting 6 days post I/R, that was attenuated in mice receiving LM11A-31. (**B**) Overall analysis of area under the curve (AUC) after 15 days of I/R experiment. Mice subjected to I/R injury showed significantly higher AUC, that was normalized in mice receiving LM11A-31. *, two-way ANOVA showed significant interaction of disease-state (ischemia) and treatment (LM11A-31) (*n* = 7–9).

**Table 1 ijms-22-00829-t001:** The sequence of the PCR primers used in the mRNA quantification experiments.

Primer	Sequence (5′–3’)
Mouse *p75^NTR^* F	CCTGCCTGGACAGTGTTACG
Mouse *p75^NTR^* R	CACACAGGGAGCGGACATAC
Mouse *VEGFA* F	GTACCTCCACCATGCCAAGT
Mouse *VEGFA* R	GCATTCACATCTGCTGTGCT
Mouse *Akt1* F	ATGAACGACGTAGCCATTGTG
Mouse *Akt1* R	TTGTAGCCAATAAAGGTGCCAT
Mouse *Bcl-2* F	GTGGTGGAGGAACTCTTCA
Mouse *Bcl-2* R	GTTCCACAAAGGCATCCCAG
Mouse *Bax* F	AGCAAACTGGTGCTCAAGGC
Mouse *Bax* R	CCACAAAGATGGTCACTGTC
Mouse *Caspase-3* F	CCTCAGAGAGACATTCATGG
Mouse *Caspase-3* R	GCAGTAGTCGCCTCTGAAGA
Mouse *P53* F	GGAAATTTGTATCCCGAGTATCTG
Mouse *P53* R	GTCTTCCAGTGTGATGATGGTAA
Mouse *GAPDH* F	CACATTGGGGGTAGGAACAC
Mouse *GAPDH* R	AACTTTGGGCATTGTGGAAGG

## Data Availability

The datasets generated and/or analyzed during the current study are available from the corresponding author on reasonable request.

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
