# Peer review of "Modulation of p75NTR on Mesenchymal Stem Cells Increases Their Vascular Protection in Retinal Ischemia-Reperfusion Mouse Model"

_ijms, 2021, doi:10.3390/ijms22020829_

Round 1
Reviewer 1 Report
Congratulations, accept in present form.
Author Response
We would like to thank the reviewer for taking the time and efforts to review our manuscript.
Reviewer 2 Report
The study of Elshaer et al entitled “Modulation of p75NTR on mesenchymal stem cells increases their vascular protection in retinal ischemia-reperfusion mouse model” is a very interesting work in the field of ischemic retinopathies. Furthermore, this study is innovative in its concept. Although there are no obvious major issues with this work, some minor issues (below) have been raised.
- The resolution and quality of all figures are poor. May be this is due to uploading issues.
- Figure 1, the micrographs do not match with statistical analysis. It says “Bright field imaging showed an increased number of acellular capillaries (red arrows) in WT that was attenuated in p75NTR-/- mice” but the number of red arrows in p75NTR-/- mice (7 arrows) more than in WT (5 arrows). Please reconcile.
- Figure 2 shows beautifully how mesenchymal stem cells incorporate to ischemic retinal vasculature after one week in Panel B (left), but surprisingly they do not differentiate into endothelial cells as they did not stain red with isolectin panel B (right). What is the difference between right and left pictures in Panel B? the same question applies to Panel A.
- Same question for Figure 3A, why mesenchymal stem cells incorporated in vasculature but did not stain with isolectin. Also, the interpretation of Figure 3A is not clear. Does scrambled-p75NTR-MSCs behave as regular MSCs? If so why there is no green or yellow color in the retinas of Scr-MSCs-injected mice as in Figure1 A or B. After how many days these photos were taken? If these cells do not behave as WT, does the method of transfection modify their homing potentiality? Please add more clarification.
- Figure 4, it is unclear why authors opted to use Western blot analysis for culture media, instead of using the ELISA method which provides more quantitative and accurate measurements.
- The rationale for Figure5, is not clear, particularly for measuring Bcl2 which is anti-Apoptotic, and BAX which is Pro- Apoptotic, and both were increased. Measuring Cytochrome C release a more robust readout. Likewise, measuring Akt mRNA expression does not mean Akt activation, measuring phospho Akt will be a more robust readout. Please discuss.
- Figure 6A and 8B, the resolution is very poor and what is the blue area in the picture denote?
- In Figures 6B and 8B, more information regarding the measurement of tube formation is needed. For example, how big the size of the area from which the measurements were taken. Are they taken from the whole well in a 24-well plate, which can be done with EVOS microscopy, or just a selected area of predetermined size? Also why highlighted tube and branching points do not cover all formed tubes in regular media treated or lipofectamine+MSc CM? The number of branching points in pictures does not reflect the number in statistical analysis.
- More clarification in the discussion is needed regarding how silencing p75NTR on mesenchymal stem cells enhances their differentiation into endothelial cells without ligand.
- It would be of great interest if the concept can be confirmed in vitro of transforming MSC into a tube-like structure using fibroblast feeder cells to establish a niche, then Knockout p75NTR, and measure CD31 expression.
- Few typos such as incorportate (line 127); cappilaries (Line 80,118, 265); v.asucular (Line 107); incorpotation (147); protein (line 170); mirographs (line 137); decreasees (line 301); effectiviness; potentia (332); invovle (359).
Author Response
We would like to thank the reviewer for taking the time and efforts to provide constructive comments and we hope that the revised manuscript will be accepted for publication. Please find answers/ explanations for the raised minor issues.
1- The resolution and quality of all figures are poor. May be this is due to uploading issues.
We agree that compression of the files adversely affected the quality of figures. Higher resolution figures are uploaded now.
2- Figure 1, the micrographs do not match with statistical analysis. It says “Bright field imaging showed an increased number of acellular capillaries (red arrows) in WT that was attenuated in p75NTR-/- mice” but the number of red arrows in p75NTR-/- mice (7 arrows) more than in WT (5 arrows). Please reconcile.
Additional representatives for both ischemic retinas of WT showing 8 acellular capillaries and p75NTR-/- showing 3 acellular capillaries re included in the revised version.
3- Figure 2 shows beautifully how MSC incorporate to ischemic retinal vasculature after 1-week in Panel-B. What is the difference between right and left pictures in Panel-B? the same question applies to Panel-A.
In figure 2, we tried to localize injected GFP-MSCs into ischemic WT retinal vasculature at two different time points; 3-days (A: upper panel) and 1-week (B: lower panel) post MSCs injection. Left and right images for each time point refer to different location in the investigated ischemic WT retinas.
But, MSC surprisingly do not differentiate into endothelial cells as they did not stain red with isolectin panel B (right). Our goal was to explore the possible behavior of MSCs post injection and its interaction with retinal microvasculature and we elected to inject large number of cells (100,000 cells/eye). At 3-days’ time-point, MSCs morphed into a satellite shape, remained green as they were not with the same plane of retinal capillaries (A: left). Other MSCs morphed into yellow ameboid shape and co-stained with isolectin (A: right). Interestingly, 1-week post injection, some MSCs integrated into retinal capillaries but remained green (B: Left) while others remained ameboid shape that co-stained with isolectin (B: Right). These observations raise the possibility of MSCs being differentiated into perivascular cells such as pericytes. Pericytes have mesenchymal origin and are traditionally recruited to support endothelial tubes [1]. Our findings lend further support to potential differentiation of injected MSCs into pericytes as illustrated previously [2,3].
4- Same question for Figure 3A, why MSCs incorporated in vasculature but did not stain with isolectin. Also, the interpretation of Figure 3A is not clear. Please refer to answer to the previous question, where it is suggested that MSCs might be differentiated to perivascular cells i.e pericytes.
Does scrambled-p75NTR-MSCs behave as regular MSCs? If so, why there is no green or yellow color in the retinas of Scr-MSCs-injected mice as in Figure1 A or B. After how many days these photos were taken? - Images were taken 1-week post MSCs injection. Scrambled-p75NTR-MSCs behaved more or less similar to the regular WT-MSCs and a representative image is now shown in the revised version with similar magnification to p75siRNA-MSC showing green cells around retinal capillaries. .
5-Figure 4, it is unclear why authors opted to use Western blot analysis for culture media, instead of using the ELISA method which provides more quantitative and accurate measurements.
We agree with the reviewer for superiority of ELISA technique for detection of trophic factors, however Western Blot (WB) is more specific for NGF as ELISA does not differentiate NGF from its precursor proNGF (See our previous publication [4-6]. Therefore, we used Western blot, which was already well-optimized and we had all required resources including antibodies for SDF1-a, VEGF-A, and NGF. Also, we were able to detect changes up to 6-Fold using WB analysis, thus we were able to prioritize detection of NGF versus proNGF and at the same time detect the relatively large quantitative difference.
6-The rationale for Figure5, is not clear, particularly for measuring Bcl2 which is anti-Apoptotic, and BAX which is Pro- Apoptotic, and both were increased. Measuring Cytochrome C release a more robust readout. Likewise, measuring Akt mRNA expression does not mean Akt activation, measuring phospho Akt will be a more robust readout. Please discuss.
We agree with the reviewer and we have updated the discussion to include these good points, including the distinction between Akt mRNA levels and actual AKT activation.
Here, we explored changes in gene expression of both survival and apoptotic markers in HREs upon treatment with CM of p75NTR-silenced MSCs. Interestingly, we’ve detected increased gene expression of both Bcl2 and Bax in treated HREs (Updated Figure 5C and D respectively). Bcl-2/Bax axis regulates mitochondrial-induced apoptosis in endothelial cells through reciprocation of the anti-apoptotic protein Bcl-2 and the pro-apoptotic protein Bax [7]. One way to interpret these results that the net effect of Bcl-2/Bax ratio was neutral on observed angiogenic behavior of HREs in-vitro as illustrated previously [7]. Also, our data show that the anti-apoptotic marker Akt and BCL2 were increased. Although the pro-apoptotic marker BAX was also increased, other pro-apoptotic markers such as P53 and Caspase3 did not show a significant difference (Updated Figure 5E and F). Since we did not find a clear and dramatic inhibitory effect on apoptosis of HREC, we did not analyze further assays such as measuring phospho Akt. However, our PCR data still shows that gene expression of the angiogenic marker; VEGFA was significantly increased in HREs, which supports enhanced angiogenesis potential of endothelial cells as illustrated in Figure 6.
7- Figure 6A and 8B, the resolution is very poor and what is the blue area in the picture denote?
Better resolution images are uploaded. The blue area indicates the baseline of plated HRE cells. Before taking images of migration, we manually made a baseline marker under the culture plate with a blue marker pen. Next, we removed cells using the cell scraper following the baseline. The blue marks in Figure 6A and 8A are those baselines under the culture plate. Through the unique pattern of ink in the baselines, we can take a picture in the exact same area to accurately determine the distance of cell migration.
8-In Figures 6B and 8B, more information regarding the measurement of tube formation is needed. For example, how big the size of the area from which the measurements were taken. Are they taken from the whole well in a 24-well plate, which can be done with EVOS microscopy, or just a selected area of predetermined size?
- Imaging of tube formation assay was mentioned and highlighted in line 511 “Cells in five replicate fields of triplicate wells were digitally photographed with an EVOS microscope (Thermo, Waltham, MA, USA).
Also, why highlighted tube and branching points do not cover all formed tubes in regular media treated or lipofectamine+MSc CM? The images for Tube formation assay were analyzed using Angiogenesis analyzer for ImageJ [8,9]). 1.8 mm2 of area from each well were analyzed with imageJ ver. 1.53c, and 3 wells were analyzed per group. The software automatically generates the skeletonized trees overlay to highlight junction, branches, and segments (Figure 6C lower panel, 8C lower panel). Master junctions were identified as junctions linking at least three master segments.
The number of branching points in pictures does not reflect the number in statistical analysis.
The representative image is single field, whereas, statistical analysis represents the average of three different wells, five random fields per well. Method section is updated.
9-More clarification in the discussion is needed regarding how silencing p75NTR on mesenchymal stem cells enhances their differentiation into endothelial cells without ligand. It would be of great interest if the concept can be confirmed in vitro of transforming MSC into a tube-like structure using fibroblast feeder cells to establish a niche, then Knockout p75NTR, and measure CD31 expression.
The statement that was included in the original discussion “MSCs-secreted VEGF mediates the differentiation of endothelial progenitor cells into endothelial cells via paracrine mechanisms [10]” supports the proposed paracrine effect of MSC to aid endothelial cells and does not imply that MSC differentiate to endothelial cells. While we were able to observe integration of MSCs into retina vasculature, and correlate that with vascular protection post-ischemic injury yet, their exact differentiation warrants further investigation beyond the current study. The Discussion is revised to clarify that (line 377).
10-Few typos such as incorportate (line 127) Line 119; cappilaries (Line 80,118, 265) line 74, 112, 250; v.asucular (Line 107) line 102; incorpotation (147) Line 137; protein (line 170) Line 159; mirographs (line 137) Line 222; decreasees (line 301) Line 288; effectiveness Line 306; potentia (332) not found; invovle (359) Line 350. We could not locate the typos in lines indicated by the reviewer (black), instead, we located them in the lines (red). We do apologize for this oversight and these typos are corrected.
- Huber, B.; Volz, A.C.; Kluger, P.J. How do culture media influence in vitro perivascular cell behavior? Cell Biol Int 2015, 39, 1395-1407, doi:10.1002/cbin.10515.
- Xu, J.; Gong, T.; Heng, B.C.; Zhang, C.F. A systematic review: differentiation of stem cells into functional pericytes. FASEB J 2017, 31, 1775-1786, doi:10.1096/fj.201600951RRR.
- Xu, L.; Li, J.; Luo, Z.; Wu, Q.; Fan, W.; Yao, X.; Li, Q.; Yan, H.; Wang, J. Abeta inhibits mesenchymal stem cell-pericyte transition through MAPK pathway. Acta Biochim Biophys Sin (Shanghai) 2018, 50, 776-781, doi:10.1093/abbs/gmy072.
- Ali, T.K.; Al-Gayyar, M.M.; Matragoon, S.; Pillai, B.A.; Abdelsaid, M.A.; Nussbaum, J.J.; El-Remessy, A.B. Diabetes-induced peroxynitrite impairs the balance of pro-nerve growth factor and nerve growth factor, and causes neurovascular injury. Diabetologia 2011, 54, 657-668, doi:10.1007/s00125-010-1935-1.
- Mysona, B.A.; Matragoon, S.; Stephens, M.; Mohamed, I.N.; Farooq, A.; Bartasis, M.L.; Fouda, A.Y.; Shanab, A.Y.; Espinosa-Heidmann, D.G.; El-Remessy, A.B. Imbalance of the nerve growth factor and its precursor as a potential biomarker for diabetic retinopathy. Biomed Res Int 2015, 2015, 571456, doi:10.1155/2015/571456.
- Elshaer, S.L.; El-Remessy, A.B. Deletion of p75(NTR) prevents vaso-obliteration and retinal neovascularization via activation of Trk- A receptor in ischemic retinopathy model. Sci Rep 2018, 8, 12490, doi:10.1038/s41598-018-30029-0.
- Molostvov, G.; Morris, A.; Rose, P.; Basu, S. Modulation of Bcl-2 family proteins in primary endothelial cells during apoptosis. Pathophysiol Haemost Thromb 2002, 32, 85-91, doi:10.1159/000065081.
- Park, H.S.; Ashour, D.; Elsharoud, A.; Chugh, R.M.; Ismail, N.; El Andaloussi, A.; Al-Hendy, A. Towards Cell free Therapy of Premature Ovarian Insufficiency: Human Bone Marrow Mesenchymal Stem Cells Secretome Enhances Angiogenesis in Human Ovarian Microvascular Endothelial Cells. HSOA J Stem Cells Res Dev Ther 2019, 5, doi:10.24966/srdt-2060/100019.
- Carpentier, G.; Berndt, S.; Ferratge, S.; Rasband, W.; Cuendet, M.; Uzan, G.; Albanese, P. Angiogenesis Analyzer for ImageJ - A comparative morphometric analysis of "Endothelial Tube Formation Assay" and "Fibrin Bead Assay". Sci Rep 2020, 10, 11568, doi:10.1038/s41598-020-67289-8.
- Ge, Q.; Zhang, H.; Hou, J.; Wan, L.; Cheng, W.; Wang, X.; Dong, D.; Chen, C.; Xia, J.; Guo, J., et al. VEGF secreted by mesenchymal stem cells mediates the differentiation of endothelial progenitor cells into endothelial cells via paracrine mechanisms. Mol Med Rep 2018, 17, 1667-1675, doi:10.3892/mmr.2017.8059.
